# A Videogame as a Tool for Clinical Screening of Possible Vulnerability to Impulsivity and Attention Disturbances in Children

**DOI:** 10.3390/children9111652

**Published:** 2022-10-29

**Authors:** Almudena Serrano-Barroso, Juan Pedro Vargas, Estrella Diaz, Isabel M. Gómez-González, Gabriel Ruiz, Juan Carlos López

**Affiliations:** 1Departamento de Psicología Experimental, Universidad de Sevilla, 41018 Sevilla, Spain; 2Departamento de Tecnología Electrónica, ETS Ingeniería Informática, Universidad de Sevilla, 41004 Sevilla, Spain

**Keywords:** diagnostic tool, endophenotypes, vulnerability to cognitive disturbances

## Abstract

An attention disturbance is a problem that affects many school-aged children. The assessment in children is usually report-based, and as a result, controversy surrounds the diagnosis. To solve this issue, the aim of this study was to develop a new tool to detect possible attention-related problems and impulsive behavior in 4- and 5-year-old children. This tool was developed as an Android app and could be used to provide an early indicator of possible future development problems. A sample of 103 children (48 girls and 55 boys) was randomly selected from primary schools and assessed by Pinky-Piggy, a videogame application based on a classical paradigm in experimental psychology. Data from this app were compared with a Child Neuropsychological Maturity Questionnaire. The subjects displayed different patterns of response to play a very simple game called Pinky-Piggy. The application discriminated between high-responders and low responders. The results showed a relationship between these two profiles and the levels of attention and neurodevelopment in each group. The tool could identify different types of profiles and demonstrated its potential to evaluate endophenotypes to predict attentional problems related to impulsive behavior. Additionally, it required less time and fewer tests to identify possible at-risk populations, thus assisting in clinical diagnosis.

## 1. Introduction

Impulsive behavior constitutes a disabling feature that seriously affects the development of social, emotional, and cognitive aspects in children. Currently, several authors define impulsivity as a multidimensional functional disorder, characterized by problems in decision making, lack of inhibitory control or failure to consider future consequences, improper behavior, and/or unplanned actions [1,2]. Thus, impulsive behavior is a type of no reflective behavior, possibly due to poor development of executive functions, which is sensitive to exposure to several situations or contexts [3,4]. Executive functions refer to a variety of higher-level skills essential for adaptive behavior. Impairments in these functions cause serious problems in childhood [5,6,7,8]. In fact, a defective interaction between inhibitory control and reflective processes developed in executive functions might be underlying impulsivity. The Working Memory model has been directly involved in this behavior [9], playing an essential role in inhibition. In fact, inhibitory control has been suggested to be the main neuropsychological impairment [5], and subsequent problems could affect working memory in different grades, especially central executive [10,11]. In children, it is related to the inability to restrain a response to signs associated with immediate reinforcement, as these signs become attractive and irresistible stimuli [12,13,14,15,16]. Furthermore, impulsivity has been described as a symptom in several clinical disorders, from simple dysfunction to the diagnosis of attention deficit hyperactivity disorder (ADHD) or drug-abuse consumption [17,18]. 

The prefrontal lobe plays a central role in inhibitory processes that gradually emerge up to the later stages of adolescence. At the end of this period, the prefrontal cortex usually reaches a stable state of maturity [19,20,21]. This maturation process facilitates the formation of neuronal networks geared towards the development of complex behavior and executive functions, providing top-down control when behavior is guided by goals [19,20,22]. Inhibitory processes are included in these cognitive processes, and this may explain their link to attention [23]. The reflective faculty is usually located in these areas, giving them the ability of voluntary control of the action or goal-directed processes [24]. Therefore, the progressive development of inhibitory control throughout a solid executive function could reduce the impulsive response. Dysfunction in these processes could act as predisposing factors, making people much more vulnerable to specific maladaptive behavior [25]. In fact, a relatively low executive function has been reported to facilitate addictive behavior [26], as the more immediate gratifying stimuli often control behavior. Recent studies have included the prefrontal cortex as a structure directly related to controlled processes, impulse control, and regulation [27,28,29]. However, the evaluation of impulsivity and attention in humans is usually report-based, uses external observation reports, or involves self-report tests of both processes independently. As a result, controversy surrounds several clinical diagnoses [30,31,32]. For example, ADHD is often said to be over or underdiagnosed due to the high variability of the prevalence rates of the disorder [33,34]. Therefore, a new approach based on the study of vulnerability traits of cognitive disorders seems justified. 

The development of executive functions has been described as a multistage process [35,36], stating that different components could develop at different times. In childhood, it is difficult to detect possible future problems, as the system in this period is still immature. For cognitive problems, prevention is considered one of the most important issues; however, remarkably little is known about the evaluation of cognitive processes in children and its relationship with risk phenotypes. In this regard, the development of new and effective tools is necessary in order to assess and provide an early indicator of possible future development problems. In this paper, we provide a clinical pilot tool, the Pinky-Piggy videogame application, with which we aim to identify vulnerability to impulsivity and attentional problems. Based on animal behavior studies [37,38,39,40,41], the autoshaping procedure, and its application in human models, this app could help identify markers of impulsivity and attentional problems focusing on individual differences. When designing the game software, our objective was to discriminate the pattern of responses of the participants. The goal was to establish whether the autoshaping procedure could work in humans as a model to identify the same profiles that emerge in animal studies. We also wanted to gauge the accuracy of the app for assessing psychological disturbances. Therefore, in addition to 4- to 5-year-old children, a small group of patients with ADHD was also analyzed for the purpose of assessing the scope of this tool.

## 2. Materials and Methods

### 2.1. Participants

The study evaluated 103 children: 47 aged 4 (24 boys, 23 girls) and 56 aged 5 (31 boys, 25 girls). All participants had normal or corrected normal vision and no history of neurological disorders. Participants were randomly selected from four collaborating primary schools in Seville, Spain. Furthermore, 23 patients with ADHD (19 boys and 4 girls; aged 7–15) were evaluated under the same conditions as 4- to 5-year-old children. They were selected from ANFATHI, an association of ADHD patients. The inclusion criteria were as follows: (1) ADHD disorder diagnosed by a public health service medical psychiatrist. (2) Show no initial clinical signs of associated or comorbid disorders. (3) Age: young people aged 7–16 years. (4) Pharmacological treatment: that is, whether they are under medication. We selected participants without treatment and participants with psychostimulant treatments, excluding those participants with other treatments such as antidepressants or neuroleptics. (5) It was mandatory for parents or legal guardians to give their informed consent to participate in the project.

The study was approved by the Comité Coordinador de Ética de la Investigación Biomédica de Andalucía, Junta de Andalucía (Spain) with code 1221-N-17. Participation in the study was voluntary, and the parents gave their informed and signed consent. 

### 2.2. Measures

#### 2.2.1. Clinical Instruments

Our descriptive study compared the scores on the Child Neuropsychological Maturity Questionnaire (CUMANIN, TEA) and the types of responses in the Pinky-Piggy videogame application for children ages 4 to 5. ADHD patients were evaluated using the Child Neuropsychological Mature Questionnaire (CUMANES, TEA), a similar test designed for children over the age of seven years. These scales have been used in psychological and neuropsychological screening, especially when there is any suspicion of behavioral or cognitive difficulty. Thus, CUMANING and CUMANES are oriented to detect developmental impairments related to, among others, the prefrontal cortex, allowing for a quantitative classification of focused attention and neurodevelopment. There is no English version of this test, although there are many equivalent scales. For example, the *d2 Test of Attention* measures selective and sustained attention as the clinical instrument used in this study. With regard to neurodevelopment, there is also a large number of neuropsychological scales, although with some modifications, the Cambridge Neuropsychological Test Automated Battery (CANTAB) measures similar areas to obtain a cognitive development score [42].

Neurodevelopment level was assessed with the CUMANIN test, which evaluates different areas of neurological maturation considering chronological age. This test is standardized with high reliability (Cronbach’s alpha = 0.90 and 0.92 for 4- and 5-year-old children, respectively). It consists of 121 items grouped into seven quantitative scales, including attention, visual perception, motor function, language, and laterality. We administered the test corresponding to the following scales: psychomotor function, which measured the degree of psychomotor development by including tests of cerebellar–vestibular control; articulatory, expressive, and comprehensive language; spatial structuring, based on the knowledge of spatial notions valued in static and dynamic positions; visual perception, which consisted of the graphical reproduction of geometric figures, with increasing complexity in each trial; assessment of semantic and iconic memory by recall of each item; attention, focusing on choosing a target with a high number of distractors; and rhythm, based on the reproduction of several rhythmic series of increasing difficulty. Likewise, CUMANES consists of 12 subscales grouped in 6 areas: language, visual perception, executive functioning, memory, laterality, and rhythm. The scores in these areas allowed us to obtain an index of neuropsychological maturation as well. 

#### 2.2.2. Design of the Pinky-Piggy Videogame Application

An attractive and colorful design was chosen for the application. The game was codified for the Android operating system, thereby allowing the download to be carried out on devices such as mobile phones or tablets. The videogame application was easy to play. First, a cartoon of a money bag appeared on the screen; after a few seconds, the bag began to shake, becoming more attractive, and then coins appeared. The players had to put the coins in the Pinky-Piggy bank to get the reward. The children always had access to the success bar of the coins obtained (Figure 1). All source files are available at: https://github.com/belita/Pinky-Piggy (accessed on 25 October 2022), allowing any researcher to download the compiled software, make a new compilation, or even modify it for their own goals. 

### 2.3. Procedure

#### 2.3.1. CUMANIN and CUMANES Tests

CUMANIN sessions were held from September 2016 to March 2019. The researcher engaged the participants in conversation before starting the test to make the assessment more enjoyable. We evaluated the seven scales following the guidelines of the clinical test. Each item was assessed with the criteria of success or error; that is, 1 point for each item performed correctly, and 0 point for errors. Thus, the higher the scores, the more satisfactory the neuropsychological development. On the contrary, lower scores indicated possible neuropsychological immaturity and therefore a higher risk factor of cognitive problems. Both tests were carried out individually and administered by a psychologist; they took around forty minutes and the results were expressed as percentiles. Similarly, CUMANES was assessed from December 2018 to February 2020. 

#### 2.3.2. Training Protocol of the Pinky-Piggy Application Game

Before starting, we explained to the children how to play the game. They were told that they would find several screens and that the goal of the game is to obtain points by touching the pig when the money coins appear. After that, we asked if they had understood the rules. If they had, we asked them to click the screen in the “Start” area, and the task began. There was no previous training for the game since the task was easy to play, as identified in a preliminary study. The subjects performed 14 trials, each lasting 8 s. During the intertrial interval (ITI 15 ± 5 s) they played exploding bubbles where they had to tap on the screen to keep them focused on the task. The app recorded the number of times they touched the bag and the pig. Throughout the trials, if the child touched the bag, the program emitted a sound similar to that of a bag with coins shaking. On the contrary, if the child touched the pig, the software emitted an oink sound. Both sounds made the game stimuli more attractive and salient and provided efficient feedback. The game started with the static money bag and the pig in grey. After 4 s, the bag began to shake and jingle the coins, and the pig turned pink; 4 s later, the bag disappeared, and the program displayed some coins for 2 s. In order to put the money into the Pinky-Piggy bank, the children just had to touch the pig once. Although all other touches were registered, they did not influence either the obtaining of the reward or the conditional and unconditional stimuli relationship. All time sets were selected from a previous pilot study with a reduced sample of children and based on animal models of autoshaping procedures. The game took 20 min and was administered individually by a psychologist. The data were collected and exported to an Excel file. 

#### 2.3.3. Index Score for Behavioral Classification

The population was divided into two profiles: high responders (HR) and low responders (LR). To divide the population, we added the total number of touches, and a limit was obtained as the weighted average. That is, the limit was the result of adding and subtracting the standard error from the average. Therefore, the threshold to be considered HR was to exceed an average of 4 touches per trial. Therefore, subjects with an average score above 56 touches per session were considered HR (see Formula (1)). The proposed parameter is based on the assumption of the lowest number of touches they might make in order to explore the system and obtain the reward throughout the trials.
(1)$$\frac{\text{Number}\;\text{of}\;\text{total}\;\text{clicks}}{\text{Number}\;\text{of}\;\text{trials}}=\mathrm{LR}\;\mathrm{if}<=4:\mathrm{HR}\;\mathrm{if}>4$$


Score index calculation. Participants with an average of four responses or less were classified as LR. Over the 4 responses, the participants were classified as HR. 

The children were classified into 2 groups: LR (score index ≤ 4 response average) and HR (score index > 4). In addition, the HR group was classified into two subgroups; these were HR for the pig and HR for the money bag.

Children were individually evaluated in their own schools for several days. First, they were evaluated with the CUMANIN test and then using the Pinky-Piggy app. Subsequently, they were classified as HR or LR. In addition, we asked teachers for their opinion about the probability of patterns of impulsivity or a possible trend of ADHD in each child according to their behavior at school. In the same vein, patients were evaluated at the ANFATHI center, but in this case we used the CUMANES test. 

#### 2.3.4. Data Analysis

Participants were classified using the Pinky-Piggy index score. Statistical analysis was performed using SPSS version 21.0 for Windows. The groups were compared using ANOVA to analyze the differences between HR to the piggy, HR to the bag, and LR. A *t*-test was used when only two groups were available; for example, when we studied differences by gender, age, or profiles such as HR and LR. Furthermore, we used the Pearson correlation coefficient between the CUMANIN scales and the Pinky-Piggy index score to assess the scope of the software. 

## 3. Results

### 3.1. Data Screening Process

The percentages of the profiles found in the 4- and 5-year-old children’s groups were 16.5% in the HR group to the pig and 10.7% for HR to the bag, with the remaining subjects being LR. The pooled groups indicated that the final sample was 27.2% in the HR group and 72.8% in the LR group. 

As stated above, the software recorded the responses of participants to the bag and the piggy when they tried to receive the reward (see Figure 1). Although it is a simple game, an ANOVA showed that the children played using different patterns of responses. In fact, we found that the participants followed different behavioral patterns; that is, some children touched the piggy with a high frequency, while another group of participants did the same with the bag (HR to the piggy F(2, 100) = 134.03; HR to the bag F(2, 100) = 125.40; both *p*s < 0.001, both ɳ2 > 0.72). However, the vast majority learned that it was irrelevant to touch the screen in order to receive the reward. Figure 2A,B show the performance in each group pooled in LR, HR to the bag, and HR to the piggy. In the same vein, Figure 3 displays this performance grouped by gender in the HR group. It is remarkable that we found a significantly higher percentage of HR boys than HR girls (Figure 3). Furthermore, data indicated that the number of contacts to the piggy was significantly greater in HR boys than in HR girls (t(15) = 3.07; *p* = 0.016, Cohen’s d = 1.55; Figure 3A). However, we did not find any statistical difference between HR boys and HR girls in the number of touches on the bag, probably due to the reduced number of girls in HR (t(9) = 0.689; *p* = 0.5; Figure 3B). 

### 3.2. Overall Attention Data

Regarding attention, all of the data in children aged 4–5 did not show a significant interaction between age and gender (F(1, 99) = 0.349; *p* = 0.55; Figure 4A,B). However, the analysis of gender as the main effect indicated higher scores in girls than in boys. That is, boys showed a lower level of attention than girls (F(1, 99) = 5.99; *p* = 0.016, Cohen’s d = 0.48; Figure 4B). To analyze whether attention level was correlated with the profiles recorded by the Pinky Piggy software, we studied the mean distribution (percentiles) reached by LR and HR participants in the attention score. Figure 5A shows these scores by group, revealing a main effect of Attention, since LR showed a higher percentile mean than both HR groups (F(2, 100) = 6.63; *p* = 0.002; ɳ2 = 0.11; Figure 5A). 

Regarding the HR group, we did not find any significant differences between the HR to the piggy and HR to the bag groups (*p* = 0.9); as a result, and due to the size effect being small, these data were pooled as a single HR group in order to analyze the distribution of the percentile of attention (Figure 5B). As Figure 6 shows, we observed that around 75% of participants included in the HR group did not exceed the 40th percentile. In contrast, the LR group showed a normal-like distribution, with the highest percentage of scores placed in the 40–60th percentile range (Figure 6). This pattern was similar in 4-year-old and 5-year-old children, and no significant differences were found between age and participant profile (F(1, 99) = 0.239; *p* = 0.62). Finally, although we did not find statistical differences between the groups (because of the reduced sample), the HR boys showed a lower level of attention than the HR girls (t(26) = 0.67; *p* = 0.5; Figure 7A). On the contrary, the attention scores were different between the HR and LR groups in boys (t(53) = 2.31; *p* = 0.025, Cohen’s d = 0.64; Figure 7B) and girls (t(46) = 2.24; *p* = 0.03, Cohen’s d = 0.86; Figure 7B). No differences were found between the boy and girls LR participants (t(73) = 1.62; *p* = 0.10).

### 3.3. Neurodevelopmental Data

Regarding the neurodevelopmental data obtained from the CUMANIN scales, it is important to note that the children obtained scores within the normal limits for the Spanish population. In fact, we did not find any differences between gender and age (4–5 years old; Figure 8A,B), neither in the interaction (F(1, 99) = 2.17; *p* = 0.14) nor in the main effects (both ps > 0.11). As with Attention, data were pooled into two categories: HR and LR. Statistical analysis indicated that the coefficient of development was significantly different in both groups. Therefore, LR showed higher records than HR (t(101) = 2.01; *p* = 0.04, Cohen’s d = 0.44; Figure 8C). Specifically, these differences were observed on nonverbal scales, a subgroup of scales used by CUMANIN to obtain a global record of neurodevelopment (see Table 1). On the contrary, no differences were found in the verbal subscales (all ps > 0.1). A more thorough analysis showed that these differences were due to the performance of boys in the HR group, as they had lower scores on nonverbal scales than observed in girls (t(26) = 2.10; *p* = 0.04, Cohen’s d = 0.87). However, no differences were found in verbal and nonverbal subscales between girls and boys in LR (both ps > 0.26). 

### 3.4. Teachers’ Reports

Finally, regarding teachers’ reports on the likelihood of children having attentional problems due to impulsive behavior, we found no significant statistical effects between attention scores and their teachers’ classifications (t(101) = 1,66; *p* = 0,10. Table 2). However, there was a negative correlation between HR and LR in regard to attention (r = −0.32; *p* = 0.001). In other words, the Pinky-Piggy videogame application was a better predictor of attentional problems than their teachers’ reports. This is important because teachers’ opinions are essential in the diagnosis of ADHD. 

### 3.5. Assessment of Patients with ADHD Diagnosis 

In order to evaluate whether Pinky-Piggy software could be sensitive to people with attentional problems, a small cohort of ADHD patients agreed to our request to participate in the assessment. We evaluated 23 children/adolescents, 5 participants without any medication, and the rest under stimulant treatment, mainly methylphenidate. These doses ranged from 20 to 70 mg depending on the severity of the symptoms. Although we reported this information, we did not include it as a variable due to the reduced number of participants. Thus, we divided the group into two subgroups, one of them named under medication and the other without medication. Despite this difference, all participants showed a medium or medium-low level of sustained attention. 

Figure 9A shows the performance of both groups in the Pinky-Piggy game. The no medication group displayed a higher number of clicks on the bag, the most prominent stimulus, and they did not pay attention to the piggy (t(21) = 3.46; *p* = 0.002, Cohen’s d = 1.7). In contrast, the mean of the under medication group was slightly higher to the piggy. This was due to two different ways to play the game. As in children aged 4–5 years, we found two populations, the HR and LR. Figure 9B shows this different performance. HR mainly pressed the piggy and was significantly higher than LR (t(16) = 3.30; *p* = 0.004, Cohen’s d = 1.59; Figure 9B). 

## 4. Discussion

The results of the present study show that the participants adopted different patterns of responses to play an essentially simple game such as Pinky-Piggy. A group of participants used fewer contacts to obtain the reward, regardless of the shift in the stimuli’s incentive salience. Another group clicked on the image of the pig when it changed color. A final group of high responders touched the bag when it began to shake. The participants were divided into two groups: high and low responders. This study shows the relationship between these two profiles of behavior and the levels of attention and neurodevelopment in each group.

Early detection of vulnerable traits in childhood is one of the most important issues for identifying possible mental disorders in adolescence and adulthood. Current evaluation in children is not targeted to detect the risk phenotype before symptoms appear but rather when relatives or teachers consider child behavior inappropriate or maladaptive [43]. Thus, prevention is underestimated and makes future cognitive problems stronger. Our study has tried to develop new strategies to detect a possible risk factor in 4- to 5-year-old children, that is, possible future cognitive development problems before symptoms appear. Based on autoshaping, an experimental psychology procedure, we created a game app to quantify traits in an objective manner. The results showed that the app allowed us to detect difficulties in executive functions associated with the frontal lobe, such as attention and possible impulsive behavior, in a sample of pupils. The level of precision was higher than the teachers’ estimation. Interestingly, these traits were also observed in a cohort of adolescents with ADHD. Therefore, the importance of detecting a propensity toward developmental difficulties lies in the possibilities of finding a better treatment.

The study of attention has been closely linked to impulsivity. In fact, DSM-5 includes ADHD disorder as one of the diagnostic conditions. Therefore, tools capable of facilitating the detection of at-risk populations could prove to be highly useful. Here, we found a possible behavioral pattern that we named LR and HR. The relationship between low levels of attention and this behavioral pattern, or the type of response observed in the 4-year-old and 5-year-old, might be a sign of potential cognitive problems. For example, the kind of response analysis regarding attention could be related to an abnormal development of this cognitive characteristic in this age group. This fact is remarkable in the population with a diagnosis of ADHD. As the results showed, participants without medical treatment displayed a high score in the app (HR profile), specifically for the most salient stimulus. However, the medication group showed two types of participants, the LR and HR. The different performance in this group may be due to medication. The response to treatment in the LR group could be more effective than in the HR group. This effectiveness could decrease the activity in ADHD and consequently reduce the number of responses to the app, although this is only an idea for future studies.

These different responses are also observed in animal models when the incentive salience of a conditional stimulus changes. The sign trackers displayed a similar pattern of behavior to the HR participants in the present study. Therefore, the higher the number of responses, the greater the probability of displaying a lower score on the attention tests. Measurement of a behavioral trait by an index could help determine individual variations to attribute incentive salience to conditioned cues [44,45,46], and this index could work as an indicator of vulnerability to impulsive behavior and low levels of attention [47]. The development of objective tools based on experimental procedures tested in animals could help us to understand the factors underlying these disturbances [24,48]. We propose the use of the Pinky-Piggy application game as a tool to predict possible patterns of impulsive responses related to low levels of attention. Impulsive behavior associated with attentional problems is a most disabling feature, and its negative effect on children’s academic development fully justifies an early and adequate evaluation [49,50].

### Possibilities of the App

The study of video game applications for diagnosis and treatment is not new. In this regard, Peñuelas-Calvo et al. [51] have reported a comprehensive review of the game tools used for the diagnosis of ADHD. The diagnosis based on games is a quite interesting approach because it allows us to have greater involvement of the patient, as they could forget that they are under evaluation. Similarly, the Pinky-Piggy videogame application was developed for use on a tablet for greater accessibility. This tool was not intended to assess children with ADHD, but rather as a fast screening for children suspected of future attention problems. Additionally, it is easy to use and only takes 20 min to prepare the setup and test. Therefore, it is simple and enjoyable for children to play. Training can take place in varying locations, such as hospitals, private centers, schools, etc. The application can be run anywhere, and there is no need for an internet connection during testing. In the future, the validation of the test could be adjusted to accommodate different ages using different levels of difficulty.

Implementing this new digital tool could facilitate a more objective classification system for attentional and impulsive behavioral disturbances. It was made to tackle the lack of fast screening tests in early detection. Furthermore, these tools are expected to boost information exchange between researchers and clinicians. The tools described here are designed to facilitate the digitization of information, leading to swift visualization and data analysis globally. This kind of procedure could be of great assistance in the early stages of clinical diagnosis. Our results suggest that it is necessary to validate this game in a wider population of children. In fact, research is already underway to extend it to all ages in primary school, including populations with ADHD diagnosis. Although this estimation model needs to be validated to obtain definitive conclusions that may lead to clinical decision making, our results are consistent with previous studies in animal models [52,53]. In this study, we have provided further evidence that our videogame application was able to identify different kinds of performance regarding the autoshaping procedure and it would work as a model for evaluating possible endophenotypes in the future.

Our final point concerns the criteria for identifying patterns of attentional problems due to impulsive behavior. When we compared teachers’ judgments with the results of a validated attentional test, we found no accurate early detection of attentional problems. Yet, teachers’ judgment is one of the factors taken into account in the diagnosis of ADHD. This might explain the overdiagnosis of ADHD: the lack of homogeneous criteria for assessment could underlie a lack of accuracy when it comes to diagnosis.

Due to the fact that the app is a new tool, the present study had several limitations that must be taken into account. First, the data from the participants should be followed in the future to analyze whether the tool is correct. Our cross-sectional study design does not allow the establishment of clear relationships between variables and risk traits. We are conducting a follow-up study of the children to conclude whether the HR group was a risk group; however, it is still early to analyze whether the tool shows enough predictive validity. In addition, the ADHD group needs to expand and try to replicate our findings in a clinical setting in which ADHD should be divided into types according to diagnoses. Although our aim was to search for vulnerability traits, we never associated the scores with the diagnosis of ADHD in pupils, as ADHD diagnoses in young people could stigmatize them. In this regard, this tool could never be used to diagnose ADHD but rather to only analyze possible risk factors. Future research could help detect a possible risk population, thereby avoiding the unjustifiable high number of false positives. This would also reduce drug consumption, specifically in patients before the age of 18, when cortical maturity is not fully complete yet. 

## 5. Conclusions

This study highlights the importance of objective tests for early detection. Few tools are capable of evaluating behavioral patterns to predict a propensity to developmental difficulties. Although our study presented descriptive findings, it was also part of an attempt to design an objective test to facilitate early detection of risk patterns of possible future disease that could lead to better treatment. While these results need to be validated for definitive conclusions, there is little doubt about the potential of digital tools to support clinical interpretation and improve efficacy in both evaluation and pharmacological and therapeutic approaches.

## Figures and Tables

**Figure 1 children-09-01652-f001:**
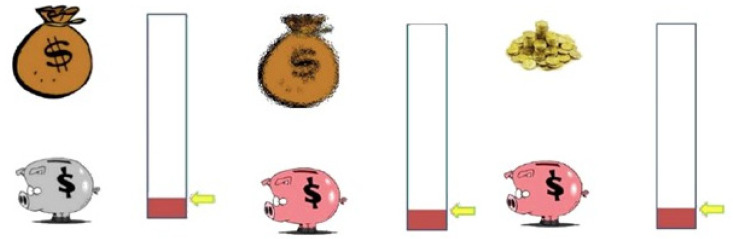
The Pinky-Piggy videogame application is based on autoshaping, a classical experimental procedure in animal models. The CS (a lever in the rat procedure) is turned into a colorful and attractive pig and money bag. The bag starts shaking after 4 s, becoming more noticeable. The US (money coins) is shown after the bag presentation. For each trial, the CS is displayed for 8 s through 14 trials.

**Figure 2 children-09-01652-f002:**
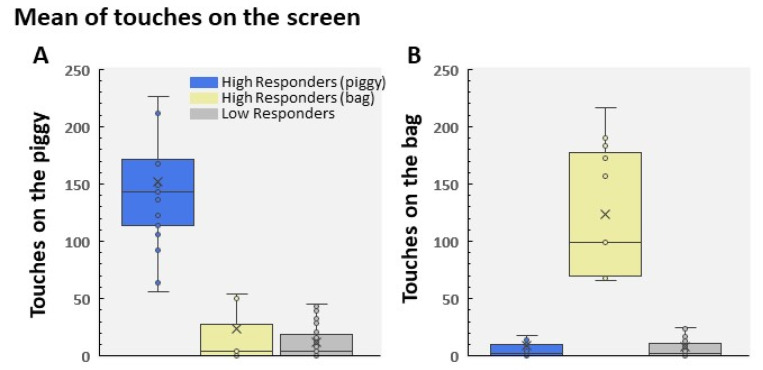
Mean of touches to the piggy (**A**) and to the bag (**B**). The sample was divided into LR (n = 75) and HR (n = 28), as indicated by Formula (1) in the Index Score for Behavioral Classification section. Furthermore, HR participants were assigned to two subgroups: HR to pig (n = 17) and HR to bag (n = 11). The graphs show the different performances followed by the participants. Only 27.2% were considered HR (16.5% to the piggy and 10.7% to the bag); the rest were considered LR. The different performances shown in A and B were statistically significant (both F(2, 102) > 114; both *p*s < 0.01).

**Figure 3 children-09-01652-f003:**
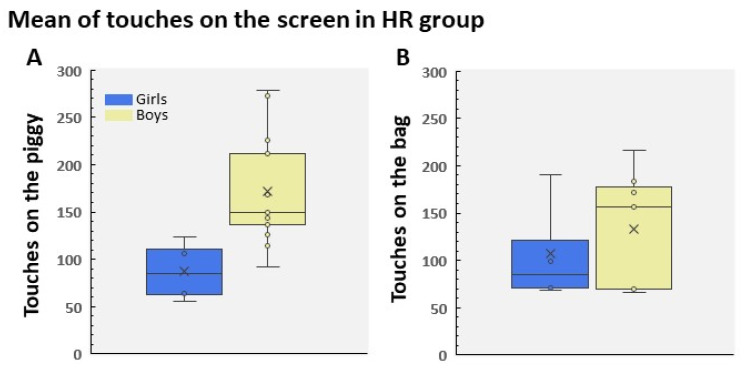
Mean of touches to the piggy (**A**) and to the bag (**B**) in the HR group. The number of participants in each group was 20 boys and 8 girls. The distribution by gender was as follows: HR to the pig (boys n = 13; girls n = 4) and HR to the bag (boys n = 7; girls n = 4).

**Figure 4 children-09-01652-f004:**
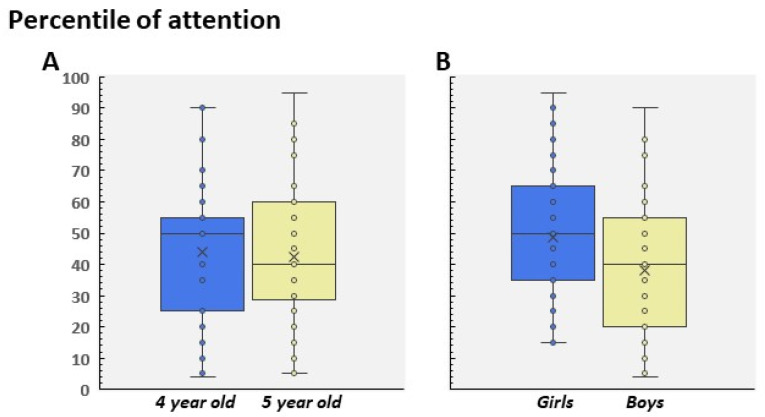
(**A**) Mean percentile of attention scored in 4- and 5-year-old children. (**B**) Mean percentile distributed in girls and boys.

**Figure 5 children-09-01652-f005:**
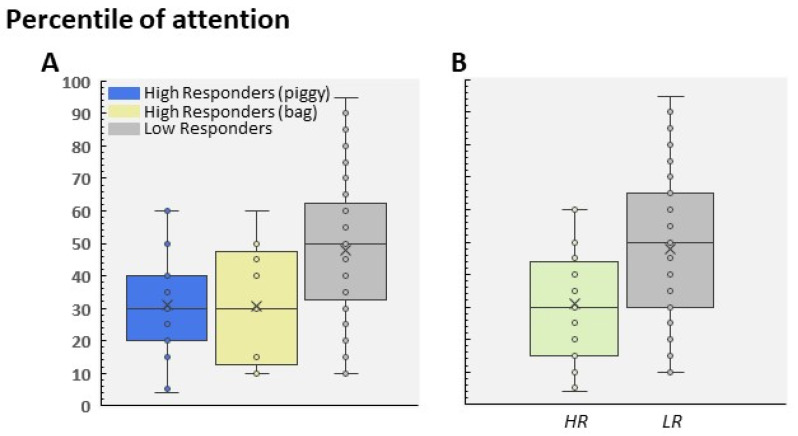
(**A**) Mean percentile of attention distributed by HR to the piggy, to the bag, and LR. (**B**) Score in the LR and pooled HR groups.

**Figure 6 children-09-01652-f006:**
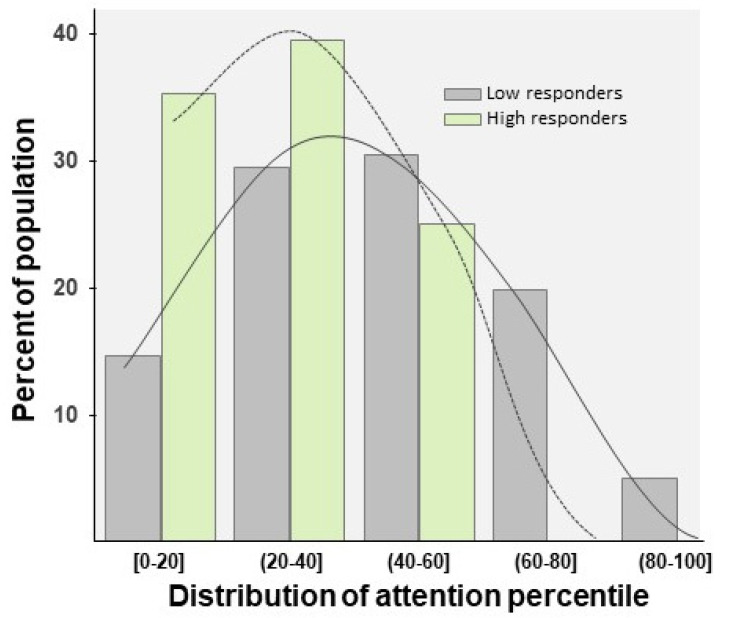
The graph shows the distribution of the LR and HR population according to the percentile of attention. Dotted line: HR scores; solid line: LR scores.

**Figure 7 children-09-01652-f007:**
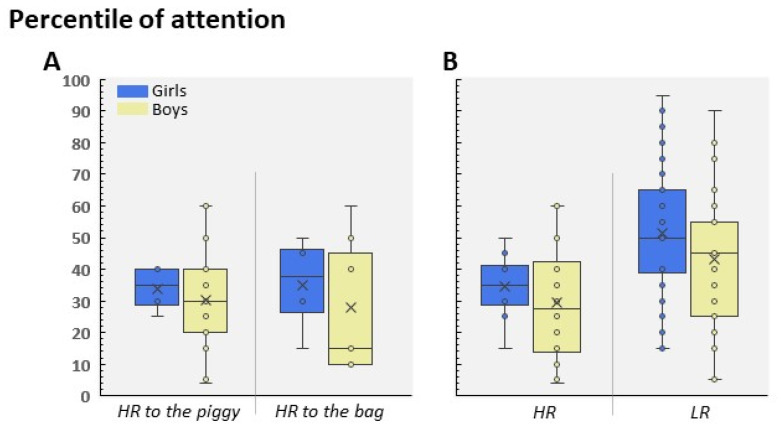
(**A**) Mean of gender-specific attention scores in the HR groups to the piggy and bag. (**B**) Distribution of attention by gender and LR and HR groups.

**Figure 8 children-09-01652-f008:**
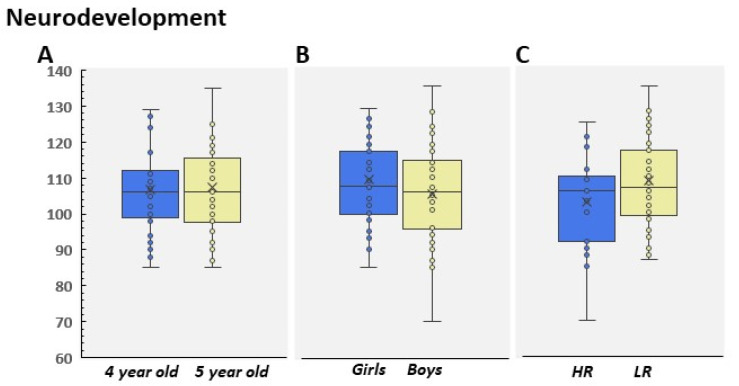
(**A**) Mean neurodevelopmental coefficient distributed by (**A**) age (4- and 5-year-old children); (**B**) gender, and (**C**) LR and HR group. HR scores are shown according to the high responder to the pig or to the bag.

**Figure 9 children-09-01652-f009:**
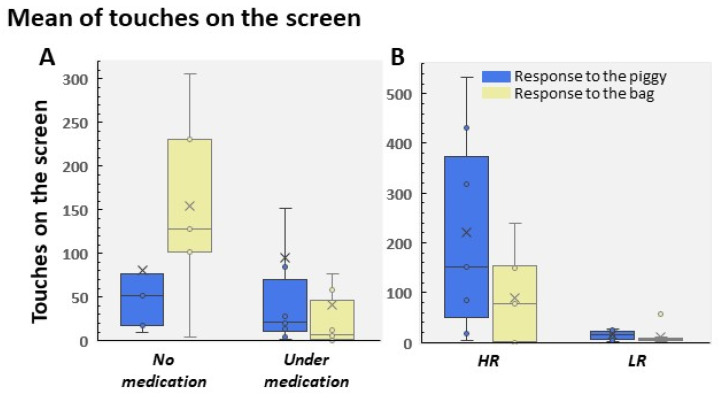
Mean of touches to the piggy and to the bag in the ADHD sample. Participants were distributed by (**A**) medical prescription and (**B**) HR/LR in patients on medical prescription.

**Table 1 children-09-01652-t001:** Results recorded in the LR and HR groups in the CUMANIN test. The areas evaluated in the table were used to reach an index of neurological maturation according to chronological age. The asterisk indicates the areas where HR showed a significantly lower score than the LR group.

	Profile	Mean	Std Error	t-Value	*p*
Psychomotor development	HR	49.2	5.1	0.944	0.34
	LR	55.01	3.2		
Articulatory language	HR	70.8	4.1	1.035	0.30
	LR	75.73	2.5		
Expressive language	HR	57.1	5.1	1.558	0.12
	LR	66.2	3.1		
Comprehensive language	HR	59.2	5.9	0.489	0.62
	LR	55.9	3.4		
Spatial structuring	HR *	59.1	6.4	2.391	0.01
	LR	74.2	3.0		
Visual perception	HR *	51.6	5.4	2.430	0.01
	LR	65.3	2.8		
Memory	HR	70.7	4.9	1.186	0.23
	LR	63.6	3.1		
Rhythm	HR	48.9	5.6	1.493	0.13
	LR	58.3	3.2		
Verbal development	HR	57.5	4.2	1.172	0.24
	LR	64.05	3.0		
Non-verbal development	HR *	56.1	5.5	2.308	0.02
	LR	69.7	2.9		

**Table 2 children-09-01652-t002:** Mean of attention scores by groups according to teachers’ opinions as a possible risk population and recording of the software of the Pinky-Piggy application videogame. The rows show the mean of attention (in percentile score) of students marked as present (attentional problems) or absent according to the teachers’ opinions. The columns display the same information under the Pinky-Piggy app. The table indicates that the software was more sensitive in discriminating between participants. In fact, the last column (total) shows a similar percentile between Present and Absent.

Teachers’ Responses	Mean of Attention Scores		
	Low Responders	High Responders	Total
Present	44.83 ± 4.22	29.05 ± 4.62	39.25 ± 3.34
Absent	49.55 ± 3.08	33.63 ± 4.26	46.36 ± 2.73
	47.60 ± 2.51	30.85 ± 3.24	

## Data Availability

Data Availability Statements in section “MDPI Research Data Policies” at https://www.mdpi.com/ethics (accessed on 25 October 2022).

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
