# Peer review of "A Videogame as a Tool for Clinical Screening of Possible Vulnerability to Impulsivity and Attention Disturbances in Children"

_children, 2022, doi:10.3390/children9111652_

Round 1

Reviewer 1 Report

Dear Authors

Some observations are presented below:

In the first instance, a general doubt that I think is necessary to request clarification is due to the title of the study. In line 60 the term "vulnerability" is mentioned for the first time. As it is considered in the title and the keywords, it is convenient to give a greater explanatory context to the fact that the lack of executive control can trigger impulsive behaviors that can have a maladaptive effect. Likewise, the term attention or attention disorder is included in the title and presents little contextual explanatory development supported bibliographically. In this, the title seems to be oriented towards the study of executive functions and impulsiveness in people with attentional difficulties. In this aspect, in materials and methods, they refer that the sample comes from ADHD Centers. Why don't they specify that they are under that context in the introduction and later in the discussion?

Therefore, the proposal to evaluate a tool that presents indicators of some of these executive functions or components of them could be incorporated in the context of the introduction as well as in the discussion and from there justify the study. On the other hand, it is necessary to take into account that the reinforcement control is more executive if it is to evaluate the proposed instrument, more reported in publications of warm executive functions.

Other minor observations:

Line 29: It is suggested to change “Impulsivity is currently defined…” for the expression “Some authors impulsivity is defined”. The cited publications refer to this concept in a particular context.

Line 32-40: It is suggested to specify more simply the relationship between impulsivity and executive functions (EF), especially working memory (WM). EFs may play a role in the anticipation of impulsive behavior, where inhibition abilities may contribute to effective performance, especially in WM.

Line 56: In addition to self-reports, include external observation reports that contain a high degree of subjectivity, which are considered background for a diagnosis.

In materials and methods.

The procedure is sufficiently explained. It is important to add the validity and reliability data of the tests used that justify their incorporation in the study, both in the executive functions test and with respect to the results obtained with the video game.

In results and discussion requires review:

Line 334-341: The suggestion is that what is observed are types of responses used rather than mentioning the use of strategies. The nature of the test in the video game reports answers, not strategies. The interpretation obeys an executive functioning that could be related to a strategy, as exposed in the text. What nature is it? What cognitive aspects are involved in it? Returning to impulsiveness, how is control of it demonstrated? Bibliographic citations can be added to support this explanation.

Line 342 -355: In this paragraph, show a relationship between the test to an explanation of executive behavior. Vulnerable in what if not defined? It is good that it is an early detection tool, but the fact that it is a vulnerability detector has not been supported. Precisely, as a procedure from experimental psychology, what is observable are behaviors and responses. How do they justify that there is a strategy behind it? How can the app detect difficulties in executive functions if it only detects responses? This could present a greater explanation in the text, both at a contextual level in the introduction and in the discussion.

Line 356 -361: This paragraph begins with attention. Attention is mentioned from what context? The apps evaluate attention responses more than executive functioning or working memory or cognitive inhibition and if it is related to impulsivity, a bibliographic citation of it could be indicated. Further explanation and bibliographic support are suggested in this regard.

Perhaps it could be interesting to evaluate if there is a relationship between the results of CUMANES and CUMANIN with respect to the apps in order to suggest that the strategy referred to in the text could exist. Attentional phenomena can contribute to executive functioning, but it is not a strategy unless key information is used to develop them; in that case, it is working memory, not attention, although the associated prefrontal association cortex is involved in both functions. This aspect should be reviewed, especially in order to the proposed title. Finally, the attribution to vulnerability is theoretical and requires this evidential support with more clarity in order to support the hypothesis proposed in this manuscript.

Line 371-383: In animal models, the relationship is fundamentally behavioral and the interpretation can suggest strategies, not the other way around. The responses attributed to the behavioral model can work with the paradigm of the incentive or reinforcement or the association for a Pavlovian conditional stimulus. It is suggested to choose one of the two for the explanation. Either it associates a neutral stimulus initially to condition itself in terms of the Pavlovian theory, or it obeys the reinforcement of the elicited behavior more typical of Skinner's operant model. It is suggested not to mix both paradigms.

The theoretical relationship they establish between impulsiveness, attention, and vulnerability, should be clearly justified based on the results obtained, thus defending the title based on the evidence.

Line 383 -434: These lines defend the apps. Good. But it requires more evidence, it is not enough that it is being taken care of and the projections can be very promising. It is suggested to include a subtitle and be included in shorter form Projections and/or strengths of the study. Somehow they make it explicit in the weaknesses, but the text of these lines is somewhat contradictory.

The same conclusions correspond but highlight the findings based on evidence and then the projected interpretation.

Reviewer 2 Report

This paper is a study on a method for judging attention disorder using a video developed by the authors. Many studies have been conducted in this field on early detection of ADHD and appropriate behavioral improvement, and as a related perspective, research has also been conducted on prediction of vulnerability. Therefore, the perspectives focused on in this study are considered to match the scope of this journal.

However, as described below, past studies have not been sufficiently referenced, and information on the CUMANIN test, which is the criterion for behavioral judgment, is lacking. 

References to past research

1 Since an excellent systematic review has already been published regarding the prediction of ADHD from using video-system, it is necessary to at least thoroughly compare this article and the research it references with this study.

Penuelas-Calvo, Inmaculada, et al. "Video games for the assessment and treatment of attention-deficit/hyperactivity disorder: a systematic review." European child & adolescent psychiatry (2020): 1-16.

In addition, as far as Table 3 of the above systematic review is concerned, this study does not include new findings for predicting ADHD.

2 Regarding the CUMANIN test, which is used as a criterion for behavioral judgment in this study, there is no article in which the readers of this journal judge the content in English. Because this study relies on the CUMANIN test, a detailed comparative study of the CUMANIN test with similar past testing methods should be conducted before publication of this study. This study did not refer to research publications that would serve as the basis for such tests, and we were unable to find any relevant literature within the scope of the reviewer's search.

Round 2

Reviewer 1 Report

Dear Authors.

I appreciate the consideration of the observations made. In this regard, some doubts still emerge:

1. Well what was added regarding vulnerability on page 2

2. Regarding what is mentioned in the concept of "attention" I still do not see clarity in what has been exposed: The title refers to "attention disorder". This can be confused with ADHD, in addition to considering the population from an Association of patients with ADHD. What the authors refer to when considering the study of Attention, without being precisely ADHD, is what is NOT supported in the literature. The similarity of the cortical areas involved does not necessarily homologate the functionality attributed to impulse control and its correlation with attention unless there is some previous evidence to support it. According to the cited reference and others that appear in Pubmed, the inhibitory processes can influence attentional processes, in order to reduce distractions. The paragraph to add should be in relation to the clarification of this idea, key in your study. Precisely in the DSM 5, one of the criteria to diagnose ADHD is impulsivity. It is a very subcortical affective process that can activate or inhibit cortical functions. Frontal executive strategies can decrease the prevalence of this function. UDS mention that they do not study ADHD, however, they consider a sample population associated with ADHD. They also suggest that video game is a clinical tool. The clinical context is the possible diagnosis of ADHD. The suggestion is i) change the title in order to describe what was formulated in the objective or specify that the evaluated population presents alterations in attention, although it comes from the aforementioned Association, they do not meet the diagnostic criteria of ADHD. However, they should present a reliable indicator of attention disturbance (not disorder) or the presence of a trait that may be in the context of attention disorder. The proposal is not necessarily to eliminate ADHD as suggested by UDS since the sample comes from that context. Only clarify the concepts and variables that are key in the study.

3. Reinforcement access control is a widely described hot executive function. How do you distinguish the strategy of "auto modeling" or "autoshaping procedure" applied to the human animal, justified in another model, such as the non-human animal? Are their mechanisms different?

4. In the response regarding what was suggested in Lines 334-341, the suggestion is that what is reported are responses or performance or another synonym, but not strategies. If they intend to relate strategies to said responses or behavior of individuals, they should support it evidentially or bibliographically. This is also in paragraphs 342 to 355.

Author Response

The text includes the reviewer concerns in bold letter and after that our response.

I appreciate the consideration of the observations made. In this regard, some doubts still emerge:

  1. Well what was added regarding vulnerability on page 2

In the Introduction section, we included this text in page 2, lines 56-61. We stated that …the progressive development of inhibitory control throughout a solid executive function could reduce the impulsive response. Dysfunction in these processes could act as predisposing factors, making people much more vulnerable to specific maladaptive behavior [25]. In fact, a relative low executive function has been reported to facilitate addictive behavior [26], since the more immediate gratifying stimuli often control behavior. In that text, we tried to introduce to the readers the vulnerability concept. In fact, as stated in the previous review, the vulnerability concept refers to the possibility of having a disorder in the future not detected in the present. We consider the study of vulnerable traits to be a good approach for studying cognitive disorder, but we are not specifying what kind of traits, we are only introducing to the reader. In the next paragraph (page 2) we indicate: In this paper, we provide a clinical pilot tool, the Pinky-Piggy videogame application, with which we aim to identify vulnerability to impulsivity and attentional problems. This is, we are measuring a response in order to correlate with performance in a standardized test (attention and neurodevelopment). The kind of response showed a significant correlation with attention results, and our interest was the use of the app for a fast screening. This means any professional (from educational or clinical field) could use it to be aware of possible future problems. Finally, if the reviewer considers, we can delete the lines 67-68.

  1. Regarding what is mentioned in the concept of "attention" I still do not see clarity in what has been exposed: The title refers to "attention disorder". This can be confused with ADHD, in addition to considering the population from an Association of patients with ADHD. What the authors refer to when considering the study of Attention, without being precisely ADHD, is what is NOT supported in the literature. The similarity of the cortical areas involved does not necessarily homologate the functionality attributed to impulse control and its correlation with attention unless there is some previous evidence to support it. According to the cited reference and others that appear in Pubmed, the inhibitory processes can influence attentional processes, in order to reduce distractions. The paragraph to add should be in relation to the clarification of this idea, key in your study. Precisely in the DSM 5, one of the criteria to diagnose ADHD is impulsivity. It is a very subcortical affective process that can activate or inhibit cortical functions. Frontal executive strategies can decrease the prevalence of this function. UDS mention that they do not study ADHD, however, they consider a sample population associated with ADHD. They also suggest that video game is a clinical tool. The clinical context is the possible diagnosis of ADHD. The suggestion is i) change the title in order to describe what was formulated in the objective or specify that the evaluated population presents alterations in attention, although it comes from the aforementioned Association, they do not meet the diagnostic criteria of ADHD. However, they should present a reliable indicator of attention disturbance (not disorder) or the presence of a trait that may be in the context of attention disorder. The proposal is not necessarily to eliminate ADHD as suggested by UDS since the sample comes from that context. Only clarify the concepts and variables that are key in the study.

We agree with the reviewer’s consideration about attention, impulsivity, and prefrontal cortex. However, we did not discuss about attention, impulsivity, and prefrontal cortex, we only show its possible relationship. In order to reach a better comprehension, we have followed the reviewer’s recommendation changing the term “disorder” by “disturbances”. In addition, if we understood well, the reviewer asked for a reliable indicator of attention disturbance. We should indicate that the aim of this study was not to evaluate the attention parameters of the children, but rather to develop a tool to detect possible future attention disturbances. Additionally, we have included a clarification when we mention the ADHD group. It is an additional group, not the main one (Line 85).

  1. Reinforcement access control is a widely described hot executive function. How do you distinguish the strategy of "auto modeling" or "autoshaping procedure" applied to the human animal, justified in another model, such as the non-human animal? Are their mechanisms different?

It is an interesting question, but unfortunately it was beyond the scope of our study. However, we will have it in mind for future studies.

  1. In the response regarding what was suggested in Lines 334-341, the suggestion is that what is reported are responses or performance or another synonym, but not strategies. If they intend to relate strategies to said responses or behavior of individuals, they should support it evidentially or bibliographically. This is also in paragraphs 342 to 355.

In order to avoid terminology confusion, we have changed “strategies” term by “pattern of response”.

Reviewer 2 Report

In the first review, I recommended reject because the necessary items were not described, but the authors expressed enough in the revised version for those points, so I changed the recommendations to accept.

Author Response

There were not any questions or concerns from reviewer 2